# Pipeline Vibration Control Using Magnetorheological Damping Clamps under Fuzzy–PID Control Algorithm

**DOI:** 10.3390/mi13040531

**Published:** 2022-03-28

**Authors:** Fei Gong, Songlin Nie, Hui Ji, Ruidong Hong, Fanglong Yin, Xiaopeng Yan

**Affiliations:** Beijing Key Laboratory of Advanced Manufacturing Technology, Beijing University of Technology, Beijing 100124, China; gongfei@emails.bjut.edu.cn (F.G.); niesonglin@bjut.edu.cn (S.N.); hongruidong@emails.bjut.edu.cn (R.H.); yfl@bjut.edu.cn (F.Y.); yanxp@bjut.edu.cn (X.Y.)

**Keywords:** hydraulic pipeline, low-frequency vibration, semi-active vibration control, MR damping clamp, fuzzy–PID

## Abstract

Aiming at the problem of low-frequency vibration of the hydraulic pipeline, a new type of semi-active damping magnetorheological (MR) damping clamp structure is designed. The structure size and material of the MR damping clamp were determined. The control model of the vibration damping system was established, and the control method combining fuzzy control and Proportional-Integral-Derivative (PID) control was used to carry out the numerical simulation, which proved that the fuzzy–PID control algorithm is effective and stable. The results show that the MR damping clamp proposed in this paper can effectively suppress the axial displacement and acceleration of the hydraulic pipeline in the excitation frequency range of 1 Hz~10 Hz. This research provides a new technical approach for low-frequency vibration control of hydraulic pipelines.

## 1. Introduction

Hydraulic systems are widely used in engineering fields and play an important role in energy transmission in civil and military industries. The hydraulic system is mainly composed of hydraulic control components, actuators, power components, auxiliary components, etc., and the hydraulic pipeline is an important energy transmission path in the entire hydraulic system [1]. The hydraulic pipeline system is mainly composed of a pipeline body, pipe joints, seals, clamps, hydraulic valves, and other structural parts. Due to the pipeline installation requirements of different industrial equipment, the layout of the hydraulic pipeline system has the characteristics of space staggered and complex, and the space between the pipe body and the pipe body and between the pipe body and the adjacent accessories is narrow. With the continuous improvement in industrial production demand, the hydraulic system develops in the direction of high speed, high pressure, and the high thrust-to-weight ratio [2,3,4,5]. For a long time, the failure of hydraulic pipeline systems caused by pipeline vibration has often occurred, and the economic loss caused is also huge. The vibration of the pipeline will cause the water hammer phenomenon, which constitutes the coupling action process of fluid motion, pressure fluctuation, and pipeline vibration, etc. This coupling action is called the fluid–structure coupling form. The fluid–structure coupling vibration of the pipeline will cause the oscillation of the industrial system, reduce the reliability of the industrial system, affect the function of the industrial system, and even cause the pipeline to burst and the industrial system to be damaged in severe cases [6,7].

Hydraulic pipeline vibration control is used to control the path of structural vibration transmission and dissipate the energy of system vibration by designing a vibration controller and developing a control algorithm. The main research contents include vibration transmission path, control parameter optimization, and control algorithm implementation [8,9]. The vibration control of hydraulic pipelines has always been paid attention by some researchers. At present, the vibration control measures for hydraulic pipelines mainly include improving the stiffness of the system, adjusting the structural damping, and eliminating the excitation force of the pipelines. According to the different mechanisms of vibration control, vibration control can be divided into passive control and active control. The high damping support clamp is a relatively effective vibration reduction and noise reduction measure in the pipeline system vibration reduction measures. The purpose of absorbing vibration energy and reducing the pipeline vibration can be achieved by employing damping and vibration reduction. When the pipeline system is in a static state, the damping pipe clamp plays a supporting role, and its vibration reduction characteristics are not shown. Especially in the resonance area, the damping and vibration reduction characteristics of the damping pipe clamp are more obvious. Bezborodov and Ulanov [10,11] proposed a calculation method for an arbitrary shaped pipeline with damping supports made of the metal rubber material. A viscous damping coefficient of the clamp was obtained based on the hysteretic loop area and dry friction. The calculation results were compared with experiments with the error less than 9%. Wei et al. [12] used the phononic crystal theory to design the vibration support clamp as a one-dimensional periodic structure composed of metal and rubber and used its band gap characteristics to realize the vibration control of the ship’s hydraulic pipeline. The new type of vibration isolator has good axial vibration damping performance within the frequency range of 600–10,000 Hz and can effectively suppress the propagation of pipeline vibration to the hull structure. Jiang et al. [13] experimentally verified the effectiveness of the pounding tuned mass damper (PTMD) system for vibration control of a submerged cylindrical pipe. Therefore, experimental results demonstrate that the PTMD system is effective and efficient to suppress the forced vibrations of the submerged cylindrical pipe at the tuned frequency and is also robust over a range of detuning frequencies. Gao et al. [14] aimed to investigate the vibration and damping characteristics of the constrained layer damping pipeline conveying fluid under elastic boundary supports. The influence of the support stiffness, the fluid velocity and pressure, the thickness and the elasticity modulus of viscoelastic, and constraining layer parameters were all considered. The results indicate that an appropriate selection of the boundary support stiffness and the viscoelastic and constraining layer parameters can obtain desirable modal properties. Li et al. [15] investigated the performance of a tuned mass particle damper (TMPD). The TMPD exhibits more significantly enhanced vibration suppression capability than a conventional particle damper. Yang et al. [16] proposed a tuning method of a pipeline vibration absorber based on anti-resonance principle for the suppression of line-spectrum vibration transmission of marine pipelines. The results show that when the vibration absorber is installed, the vibration level of the target measuring point is reduced from 97.3 dB to 90.8 dB, with a decrease of 6.5 dB. The results show that the vibration absorber tuning method based on the anti-resonance theory can effectively suppress the vibration transmission of the low-frequency line spectrum of the pipeline. Zhou et al. [17] designed a frequency-modulated dynamic vibration absorber based on the anti-resonance principle, which is simple in structure, easy to install, and can adjust the frequency to meet the vibration reduction work at different operating frequencies of the pipeline system. The frequency modulation dynamic vibration absorber has an obvious effect on the vibration reduction in the pipeline system, and the vibration reduction in the pipeline system under different working frequencies can be well solved by changing the position of the mass block. Pan et al. [18] developed an active adaptive noise control method for reducing pressure pulsation in a hydraulic system. The test results showed that the by-pass structure can be seen as a general and effective approach for fluid-borne noise cancellation with good performance and few limitations. Jiao et al. [19,20,21] discussed the vibration active control methods to reduce the vibrations of fluid power supply and piping systems. A new vibration active control method with a multilayer piezoelectric technology driven orifice valve was proposed to reduce the vibrations of the fluid pulsation. The test results showed that this method was capable of reducing the vibrations at a minimum level. For active vibration control, the control effect is closely related to the control method and algorithm.

Through the analysis of the current research status of hydraulic pipeline vibration control, it can be seen that although passive vibration control has the advantages of a clear vibration reduction mechanism, simple structure, no need for energy input and easy implementation, its control frequency range is narrow, and it can only control high-order vibration frequencies. Passive vibration control is not effective for low-frequency vibration, and the control parameters cannot change with the change in the excitation environment. Therefore, the reduction effect of passive control vibration is limited. Active vibration control has a better control effect on low frequency, strong adaptability to the external environment, can automatically follow the change of vibration characteristics, and has great design flexibility; nevertheless, there are many closed-loop control systems, and the stability of the system needs to be further improved. Compared with passive control, semi-active control can track system parameters and excitation characteristics, so that the control effect is significantly improved. At the same time, compared with active control, semi-active control does not require a large control force when the control system circuit fails. When the self-tuning vibration absorber becomes a pure passive control system, it will not bring additional harm to the hydraulic pipeline system while providing passive vibration reduction. A semi-active MR damper as a damping device has a significant advantage due its fast response, good controllability, low energy consumption, and outstanding adaptability. Recently, the vibration reduction technology of MR dampers has been developed rapidly and has been successfully applied to the vibration reduction control of some civil structures, as shown in Figure 1 [22]. Our group has carried out preliminary research on the use of MR dampers for pipeline vibration control and achieved several results [23,24,25].

This paper proposes a semi-active MR damping clamp structure which is composed of sensors, controllers, and actuators. The sensor is used to detect the vibration of the pipeline, the actuator can change the vibration response of the pipeline, and the controller can collect the signal of the sensor as required to reduce the vibration response of the pipeline. Aiming at the problems that the actual working conditions of hydraulic pipelines are complex, difficult to identify, and the control system contains parameter uncertainty and actuator time delay, an identification method based on multi-sensor information fusion is designed which realizes the identification of complex working conditions and designs a controller that is used to change the damping characteristics of the magnetorheological damping clamp in real-time. The damping properties of magnetorheological damping clamps reduce vibrations transmitted to the pipe by external excitation. The research in this paper can promote the application of MR damping clamps in the field of pipeline vibration control and has high academic research significance and engineering application value.

The paper is divided into the following sections. Section 2 describes the MR damping clamp structure and working principle. Section 3 presents the control strategy as well as the controller design. Section 4 describes the implementation of the MR damping clamp test platform and control system for pipeline low-frequency vibration control. In Section 5, the results and discussion of the experimentation are presented and, finally, in Section 6, the conclusions of this investigation are shown.

## 2. Materials and Methods

The MR damping clamp structure is shown in Figure 2, along with the structure and schematic of the magnetorheological damping clamp, which is mainly composed of five parts: pipe clamp, connecting rod, magnetorheological damper, Y-joint, and base. The pipe clamp is divided into upper and lower parts. The inner surface of the pipe clamp is provided with a rectangular groove, which can increase the friction with the pipeline and prevent the pipeline from rotating. It is fixed on the pipeline and is easy to disassemble. One end of the connecting rod is hinged with the hinged seat of the inner pipe clamp through the shaft pin to form a rotating shaft mechanism, and the other end is threadedly connected with the threaded hole at the end of the piston rod of the magnetorheological damper through the external thread, the upper end of the Y-joint is hinged with the tail wing of the lower end cover of the damper, and the lower end is connected with the countersunk seat of the lower bottom plate by threaded fitting. The length of the external thread of the Y-joint screwed into the countersunk seat can be adjusted according to the actual installation height of the pipeline; there are four U-shaped grooves on the side, and the lower base plate can be fixed on the base through T-shaped bolts. The groove in the middle is the installation groove of the digital signal processing (DSP) controller, and the DSP controller is fixed in the groove of the base through double-ended studs.

The MR damper designed in this paper adopts the working mode of shear valve type, is a built-in damping channel, and adopts the structural form of double exit rods. The damper is divided into a working cylinder and an auxiliary cylinder. The function of the auxiliary cylinder is to provide a movement space for the propeller shaft and to provide a connecting member for the installation of the damper. The working cylinder is mainly composed of a cylinder block, a piston, a coil, a sealing cover, and a propeller shaft. The gap between the piston and the cylinder is the damping gap of the magnetorheological fluid. When the piston moves relative to the cylinder, the magnetorheological fluid flows from one side of the cavity to the other side of the cavity through the damping gap due to the squeeze of the piston. At the same time, it will be shared by the piston, so that the damper will generate a damping force. The coil and iron core in the piston are magnetic fields excitation devices. When a current is applied to the coil, a magnetic field will be formed at the damping channel, which will cause the change of the shear yield strength of the magnetorheological fluid, and finally realize the control of the damping force.

As can be seen from Figure 3, at the installation position of the MR damping clamp, where φ is the included angle between the axes of the piston rods of the two magnetorheological dampers, *L_OA_* and *L_OB_* are the distances from the center of the inner tube clamp to the center of the left and right hinged seats and *L_AC_* and *L_BD_* are the distances from the center of the hinged seat of the inner pipe clamp to the center of the hinged seat of the lower end cover of the magnetorheological damper. One end of the adapter is hinged with the inner pipe clamp and the other end is connected with the piston rod of the magnetorheological damper through the threaded fitting so it can adjust the value of *L_AC_* and *L_BD_* by adjusting the length of the external thread of the adapter screwed into the threaded hole of the piston rod. *h* is the distance from the center of the tail wing hinge seat of the lower end cover of the magnetorheological damper to the lower end surface of the lower bottom plate. The end of the Y-type joint is connected with the joint seat of the lower bottom plate through threaded fitting and is fixed on the lower bottom plate so it can be adjusted by adjusting the external thread of the Y-type joint. Adjust the value of h according to the length of the threaded hole in the joint seat. *L* is the horizontal distance between the center axes of the joint seats on both sides of the lower bottom plate, and *H* is the vertical distance from the center of the pipeline rigidly fixed with the inner pipe clamp to the lower end surface of the lower bottom plate.

Figure 4 illustrates the working principle of the MR damping clamp. The fluid pulsation inside the pipe and the external excitation cause the pipe to vibrate, and the pipe vibration is transmitted to the pipe clamp structure through the shell. The displacement in any direction in the two-dimensional space is transformed into the reciprocating motion of the magnetorheological damper piston in the cylinder, which in turn causes the internal magnetorheological fluid to generate shear damping force, and the current is controlled by feedback according to the pipeline vibration excitation signal. Size and change the magnetic field environment around the magnetorheological fluid form a variable damping force and achieve the purpose of suppressing the variable frequency vibration of the pipeline.

The MR damper clamp is filled with the anti-settling micro-nano composite magnetorheological fluid developed by our research group. The magnetorheological fluid uses micro-carbonyl iron powder coated with multi-walled carbon nanotubes and nano-Fe_3_O_4_ modified by surfactant. As magnetic conductive particles, the anti-settling performance of the MR damper clamp is greatly improved. The piston and propeller shaft are made of electric soldering iron DT4, which has high permeability and low remanence, and the rest of the components are made of 45# steel. The excitation coil in the middle of the piston is a copper wire with a diameter of 1 mm. The maximum load current is 2 A. The magnetic circuit of the MR damping clamp is optimized, and the design parameters shown are shown in Table 1.

## 3. Methodology

In the hydraulic system, due to the flow pulsation of the hydraulic pump and the sudden opening of the hydraulic valve, the vibration frequency of the pipeline system is time-varying and the damping coefficient of the MR damping clamp is uncertain, which makes the controlled system have many uncertain parameters. When the vibration control method is used, such as skyhook control [26], LQG control [27,28], PID control [29,30], sliding mode control [31,32], and other methods, once the model parameters change, the control performance of the system will be degraded or even deteriorated. For such uncertain problems, other types of robust control methods, such as fuzzy control [33,34] and robust control [35,36], have emerged. This kind of control algorithm has strong robustness to model parameter uncertainty and unknown disturbance. However, fuzzy control and robust control also have some shortcomings. For example, fuzzy control needs to formulate fuzzy control rules and design fuzzy membership functions based on experience, and robust control has shortcomings in terms of convergence speed and control accuracy. For the application of fuzzy control and robust control in the field of active structural vibration control, many research directions are to use compound control strategies. Therefore, this paper adopted the control method combining fuzzy control and PID control. The key parameters of PID are adjusted online by fuzzy control, which not only has the flexibility and speed of fuzzy control, but also inherits the advantages of high precision of PID control algorithm, so as to realize high-performance tracking control of the MR damping clamp.

### 3.1. Control Model

In this paper, a nonlinear model of the vibration control system is established, and a fuzzy–PID control based on state feedback is designed based on Lyapunov stability theory. Using the Matlab/Simulink simulation platform, the performance of the fuzzy–PID control system and the performance of the uncontrolled system are simulated and compared.

Figure 5 presents the mechanical model of a semi-active vibration control system. A hydrodynamic pipeline with two degrees of freedom under the action of environmental disturbance *F*(t) ∈ *Rr* is given by Equation (1).
(1)$$\left\{\begin{array}{l} M\ddot{X}(t)+C\dot{X}(t)+KX(t)=D_{s}F(t)\\ X\left(t_{0}\right)=X_{0}\;\dot{X}\left(t_{0}\right)={\dot{X}}_{0} \end{array}\right.$$
where *X* is the pipeline displacement vector, *M* is the pipeline mass, *C* is the pipeline damping, *K* is the stiffness matrix, *D_S_* is the environmental interference location matrix, $X\left(t_{0}\right)$ is the initial displacement of the pipe, and $\dot{X}\left(t_{0}\right)$ is the initial velocity of the pipe.

In order to control the response of the pipeline system, two magnetorheological damping devices are installed in the pipeline system to control the control force *U*(*t*) provided by the device, and the corresponding position matrix is *B_s_*. The motion equation of the controlled pipeline is given by the following Equation:(2)$$M\ddot{X}(t)+C\dot{X}(t)+KX(t)=D_{s}F(t)+B_{s}U(t)$$

The two ends of the Equation (2) undergo Fourier transformation, and the kinematic equation of the pipeline is established as
(3)$$-\omega^{2}MX(\omega )+j\omega CX(\omega )+KX(\omega )=F(\omega )$$

The frequency response function of the piping system can be expressed as
(4)$$H(\omega )=\frac{U(\omega )}{F(\omega )}=\frac{1}{-\omega^{2}M+j\omega C+K}$$

It can be seen from Equation (4) that the frequency response function of the pipeline system is in the form of a complex number, which adopts the form of polar coordinates, and Equation (4) can be presented as
(5)$$\left\{\begin{array}{l} |H(\omega )|=\frac{1}{\sqrt{{\left(K-\omega^{2}M\right)}^{2}+{\left(\omega C\right)}^{2}}}\\ \phi =\arctan\frac{-\omega C}{K-\omega^{2}M} \end{array}\right.$$
where H(ω) is the amplitude-frequency characteristics of the frequency response function, and ϕ is the phase-frequency characteristics of the frequency response function.

The transfer function of the piping system is the ratio of the Laplace transform of the system response to the Laplace transform of the system excitation. The Laplace transforms of excitation *F*(*t*) and response *U*(*t*) are respectively defined as:(6)$$F(s)=\int_{0}^{+\infty }F(t)e^{-st}dt$$
(7)$$U(s)=\int_{0}^{+\infty }U(t)e^{-st}dt$$
where s=σ+jω is the plural Laplace transform variable.

The Laplace transform on both sides of Equation (2) under zero initial conditions can be expressed as
(8)$$\left(Ms^{2}+Cs+K\right)U(s)=F(s)$$

Therefore, the transfer function of the piping system can be expressed as
(9)$$H(s)=\frac{U(s)}{F(s)}=\frac{1}{Ms^{2}+Cs+K}$$

### 3.2. Control Strategy

For all kinds of linear time-invariant systems, the PID control method can satisfy the control effect and is suitable for the system with fixed control object parameters and strong nonlinear. However, in the actual working conditions, the load of the control object changes with time, and there are complex interference factors. Due to the irregularity of these disturbances, the PID control algorithm cannot make real-time adjustments according to the actual working conditions. In this study, the fuzzy controller is used to adjust PID parameters to realize the continuity of parameter adjustment. To achieve this goal, a control scheme for the MR damping clamp is established in Figure 6.

The PID controller adopts a discretized control method, and its control rate equation is shown in Equation (10).
(10)$$u(k)=(K_{P}+\Delta K_{p})e(k)+(K_{I}+\Delta K_{I})T\sum_{i=0}^{n}e(k)+\frac{\left(K_{d}+\Delta K_{d}\right)}{T}[e(k)-e(k-1)]$$
where *K_p_*, *K_I_*, and *K_d_* are the initial parameters of PID control; *K_p_*, *K_I_*, and *K_d_* are the fuzzy control real-time adjustment of PID parameters. Although the PID control algorithm has been widely used in the field of active vibration control, it also has many defects. In the general active vibration PID control system, the PID parameters are fixed in the whole control process. However, for random time-varying vibration conditions, it is difficult for conventional PID control to meet the control requirements [37]. In addition, the fuzzy control algorithm can adjust the output in real-time according to the actual working conditions and has the advantages of being suitable for nonlinear control objects and having strong robustness. However, its lack of integral terms in the fuzzy control algorithm results in poor control accuracy and insensitive control. Therefore, this paper combines the fuzzy control method with PID control theory to realize the continuity of PID parameter adjustment, and the good control function of the PID control method is used, which can make the MR damping clamp exert good adaptive characteristics. The control flow chart of the parameter fuzzy–PID control strategy is shown in Figure 7.

In this study, the parametric fuzzy–PID control strategy uses the deviation of the MR damping clamp output force from the target damping force and the rate of change of the deviation as the feedback signal and uses fuzzy rules to adjust the PID control parameters online, thus improving the self-adaptive performance of the control system under different operating conditions. The PID control strategy has the advantages of good static characteristics and dynamic performance, low computational effort, and is easy to implement through the upper computer control. According to the damping characteristics of MR damping clamps, the fuzzy rules of the design parameter for the fuzzy–PID control strategy are shown in Table 2, Table 3 and Table 4.

According to the fuzzy rules shown in the table above, when the force variation error ***e***(*t*) is large, to ensure the rapidity of the control system response, a large *k_p_* should be ensured. If *e*(*k*)**^.^***ec*(*k*) > 0 and *e*(*k*) > 0, with the *e*(*k*) gradually increasing, the tendency of the damper piston to move away from the stable position gradually increases. To reduce the error, stronger controls should be used to increase *k_i_* and reduce *k_d_*, If *e*(*k*)**^.^***ec*(*k*) > 0 and *e*(*k*) < 0, the amount of control should be reduced, larger *k_d_* adopted, and *k_i_* suppressed. When the error *e*(*k*) gradually decreases and *e*(*k*)**^.^***ec*(*k*) < 0, to ensure the response speed and stability of the system, *k_p_* and *k_d_* should take medium and reduce *k_i_*. When the error *e*(*k*) is medium sized, to ensure a fast response and to avoid excessive overshoot, the *k_p_* should be a moderate value. If *e*(*k*)**^.^***ec*(*k*) > 0 and *e*(*k*) > 0, with the *e*(*k*) gradually increasing, the intensity of control should be moderate, with moderate increases in *k_i_* while *k_d_* decreases. If *e*(*k*)**^.^***ec*(*k*) > 0 and *e*(*k*), *k_d_* should be increased and *k_i_* reduced by an appropriate amount. When the error *e*(*k*) gradually decreases and *e*(*k*)**^.^***ec*(*k*) < 0, in order to ensure the response speed and stability of the system, *k_i_* and *k_d_* should take medium. When the error *e*(*k*) is small, the system is close to steady state and a weaker control effort is used. Both of *k_p_* and *k_i_* should take as large as appropriate, *k_d_* should take medium. However, if the rate of change of the error *e*(*k*) is significant, the control parameters should be adjusted slightly to take into account the trend of the error.

The MR damping clamp parameter fuzzy–PID vibration control system was established in MATLAB/Simulink software, as shown in Figure 8.

The force tracking curves of the MR damping clamp in the *x* and *y* directions are shown in Figure 9 by the fuzzy–PID vibration control algorithm. As can be seen from Figure 8, the force tracking characteristics of the fuzzy–PID control algorithm are an overshoot phenomenon. However, when the damping force is large, the overshoot is smaller and the force tracking characteristics are better.

Measurement and analysis of vibrations are described in terms of acceleration levels, in decibels (dB). The acceleration response level due to unit excitation force is:(11)$$VAL=20\mathrm{lg}\left(\frac{a}{a_{0}}\right)=20\mathrm{lg}\left(\frac{T_{A}}{a_{0}}\right)$$
where *VAL* is the vibration acceleration level, in dB, *a* is the vibration acceleration, *a*_0_ is the reference acceleration, and *T_A_* is the absolute transfer coefficient (transfer function):(12)$$T_{A}=\left|\frac{a}{F_{0}}\right|$$
where *F*_0_ is the excitation force amplitude.

## 4. Experimental Study

The experimental part of the thesis is completed on the semi-active vibration control test bench of the fluid power pipeline. This is seen from Figure 10, including that the length of the test pipeline is 3 m, the outer diameter of the pipeline is 34 mm, and the inner diameter of the pipeline is 25 mm, and the vibration control part includes the displacement sensor, controller, MR damping clamp, and other parts. When the pipeline is vibration, the displacement sensor obtains the pipeline displacement signal, the signal acquisition system collects the displacement signal, and the controller calculates the output control signal, amplifies the control signal, and applies it to the MR damping clamp. The unique MR effect of the magnetorheological fluid (MRF) in it can generate a corresponding damping force to act on the surface of the pipeline.

The excitation force during the test is adjusted by the vibration motor through the controller; the excitation signal adopts a sine signal, the excitation amplitude value is 2 mm, and the excitation frequency range is set to 1–10 Hz. Figure 11 represents the overall structure of the test device.

The controller is the core of the magnetorheological damping tube clamp control system, and it is the key factor determining the semi-active control performance of the damping tube clamp. When the control system needs to quickly detect the change in the pipeline state, it controls the pipeline according to the pipeline vibration control law. Since a large amount of data processing needs to be performed in the control process, in order to reduce the system delay and improve the control accuracy, considering the complexity of the control algorithm, the controller of the embedded system uses a DSP with powerful data processing capability as the processor. The controller is shown in Figure 12.

## 5. Results and Discussion

When the sinusoidal excitation frequencies are 1 Hz, 5 Hz, and 10 Hz, respectively, the vibration response test results of the pipeline under fuzzy vibration control are shown in Figure 13.

Under the excitation frequency of 1–10 Hz, the statistical results of the pipeline vibration response using fuzzy–PID control are shown in Table 5.

As shown in Table 5, when fuzzy–PID control is used, the maximum peak displacement and peak acceleration of pipeline vibration under different excitation frequencies are 0.3554 mm and 3.5160 m/s^2^, respectively, and the VAL of displacement is 13.0984–23.4504 dB, VAL of acceleration is 6.0256–15.9341 dB, the range of displacement attenuation is 77.87–93.28%, and the range of acceleration attenuation is 50.03–84.03%.

In addition, according to the displacement attenuation rate of the pipeline, it was found that with the increase in the excitation frequency, the displacement attenuation rate of the pipeline decreases significantly. It can be found that the low-frequency vibration can transmit more energy relative to the high-frequency vibration. Therefore, the pipeline should try to avoid low-frequency vibration. The effect of the MR damping clamp in suppressing the low-frequency vibration of the pipeline is more obvious, which is mainly due to the velocity hysteresis of the damping force of the MR damping clamp increasing with the increase in the loading frequency. This phenomenon shows that the frequency has an obvious effect on the damping force.

## 6. Conclusions

In this paper, for the engineering problem that passive control cannot effectively suppress the low-frequency vibration of hydraulic pipelines, based on the newly developed MR damping clamp, a semi-active pipeline vibration control method is studied by adopting the fuzzy–PID control algorithm. A new type of MR damping clamp is designed and manufactured, and its performance is simulated and tested. The results show that the magnetorheological damping clamp using the fuzzy–PID control algorithm has a good control effect on the pipeline. According to Table 6, a count was made of the hydraulic pipeline vibration control methods. Based on the developed magnetorheological damping pipe clamp, a semi-active vibration control experiment was carried out on the hydrodynamic pipeline using the fuzzy–PID control algorithm. It is proved that the magnetorheological damping tube clamp can effectively suppress the vibration response of periodic excitation in the range of 1 Hz~10 Hz of the pipeline vibration frequency band.

As can be seen, the maximum cancellation of VAL of displacement is 23.4504 dB, occurring at the frequency of 1 Hz, and VAL of acceleration is 15.9341 dB, occurring at the frequency of 2 Hz. The average cancellation of VAL of displacement was over 18 dB and VAL of acceleration was also over 11 dB.

According to Table 6, it is also observed that passive control technology is not suitable for pipeline systems with a certain frequency bandwidth or frequency variation. The limitation of cost and power consumption makes active control technologies difficult to practice in practical engineering applications. In addition, there is no effective method for the ultra-low frequency vibration of the ship’s hydraulic piping system below 10 Hz. The ultra-low frequency vibration of the hydraulic pipeline is controlled, and some effective methods have been obtained. As future work, we are working on implementing the MR damping clamp structure to achieve a lighter weight and miniaturization of the structure.

## Figures and Tables

**Figure 1 micromachines-13-00531-f001:**
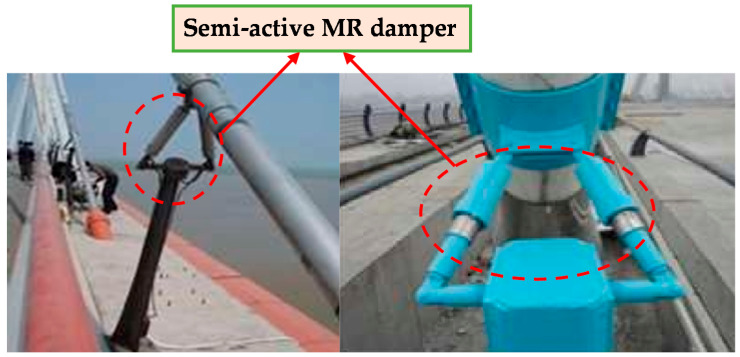
MR damper application.

**Figure 2 micromachines-13-00531-f002:**
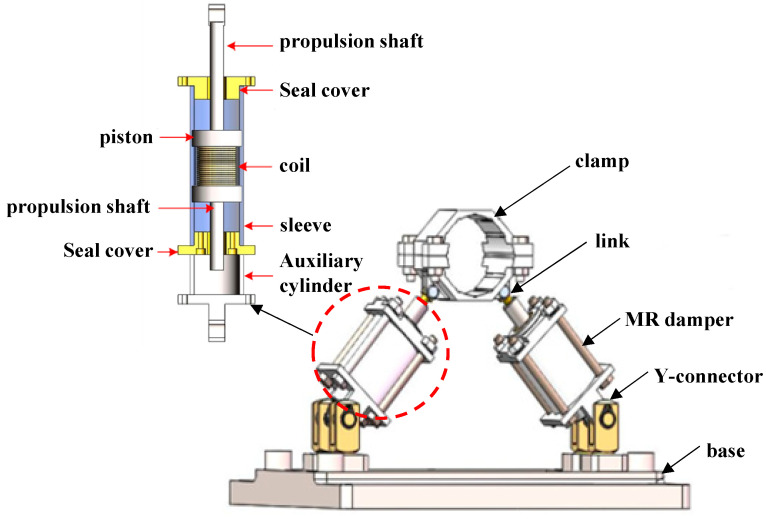
Structure and schematic of magnetorheological damping clamp.

**Figure 3 micromachines-13-00531-f003:**
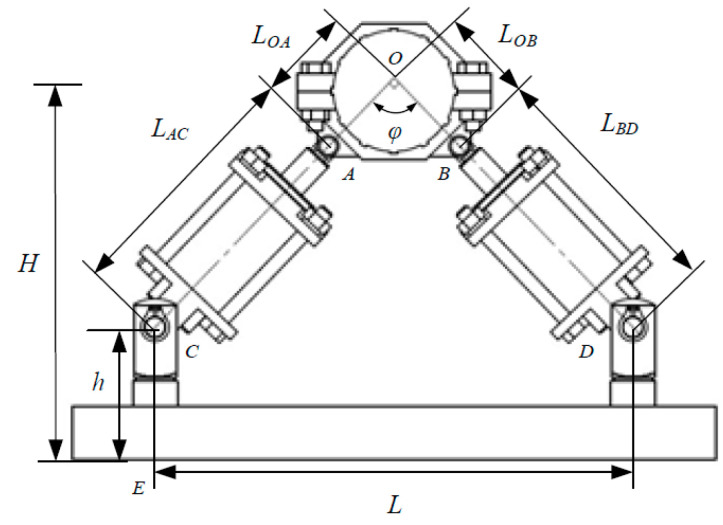
Installation position of the MR damping clamp.

**Figure 4 micromachines-13-00531-f004:**
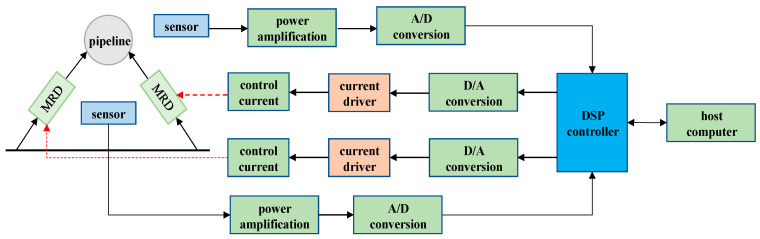
Schematic diagram of MR damping clamp.

**Figure 5 micromachines-13-00531-f005:**
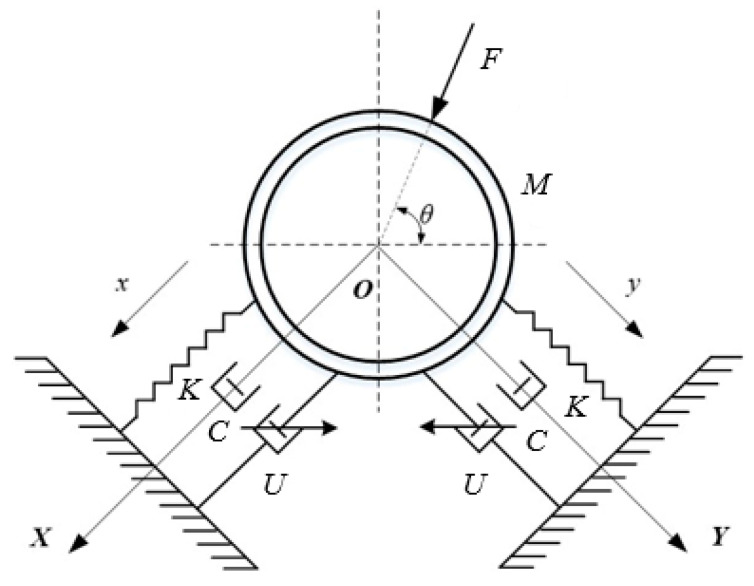
Mechanical model of semi-active vibration control system.

**Figure 6 micromachines-13-00531-f006:**
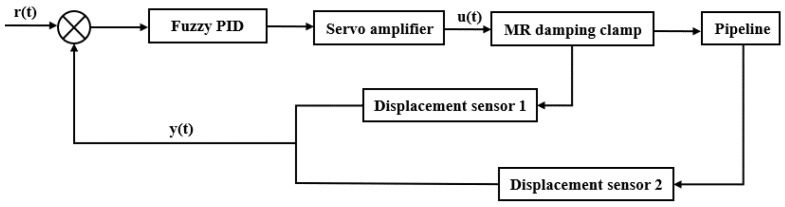
Automatic control system of the pipeline vibration.

**Figure 7 micromachines-13-00531-f007:**
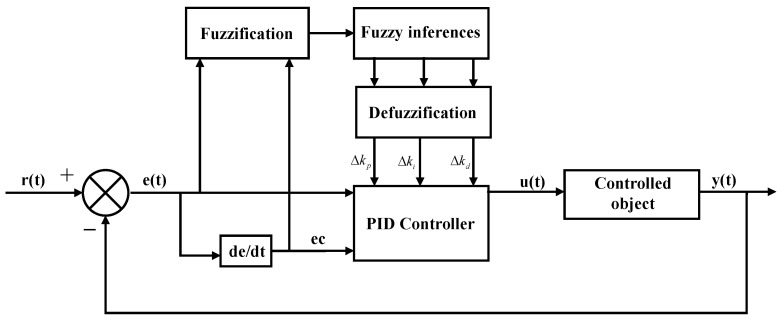
Principle of fuzzy–PID control system.

**Figure 8 micromachines-13-00531-f008:**
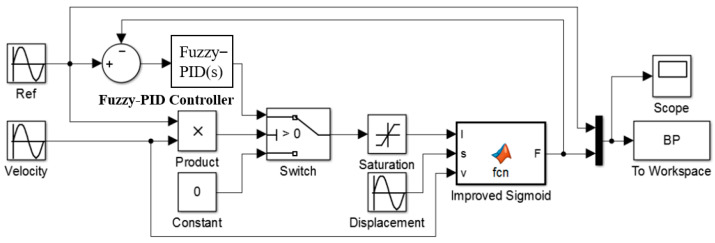
Simulink simulation model of parameter fuzzy–PID drive system.

**Figure 9 micromachines-13-00531-f009:**
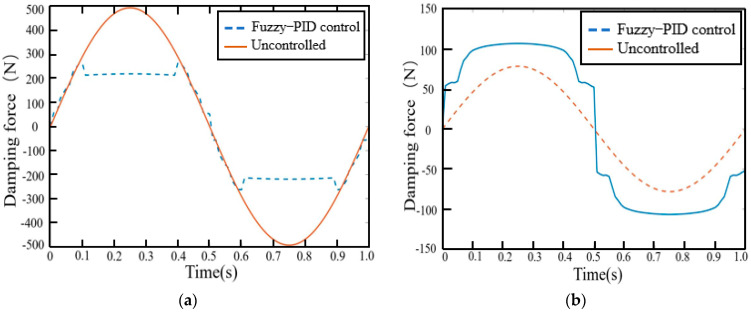
Damping force of fuzzy–PID control. (**a**) *x*-axis force tracking curve. (**b**) *y*-axis force tracking curve.

**Figure 10 micromachines-13-00531-f010:**
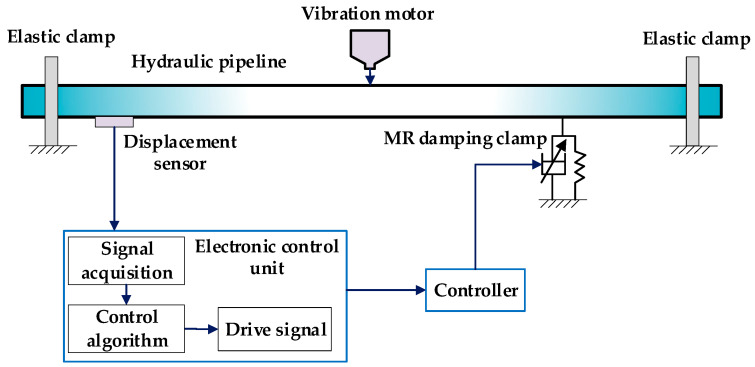
Schematic diagram of MR damper clamp pipeline vibration control system.

**Figure 11 micromachines-13-00531-f011:**
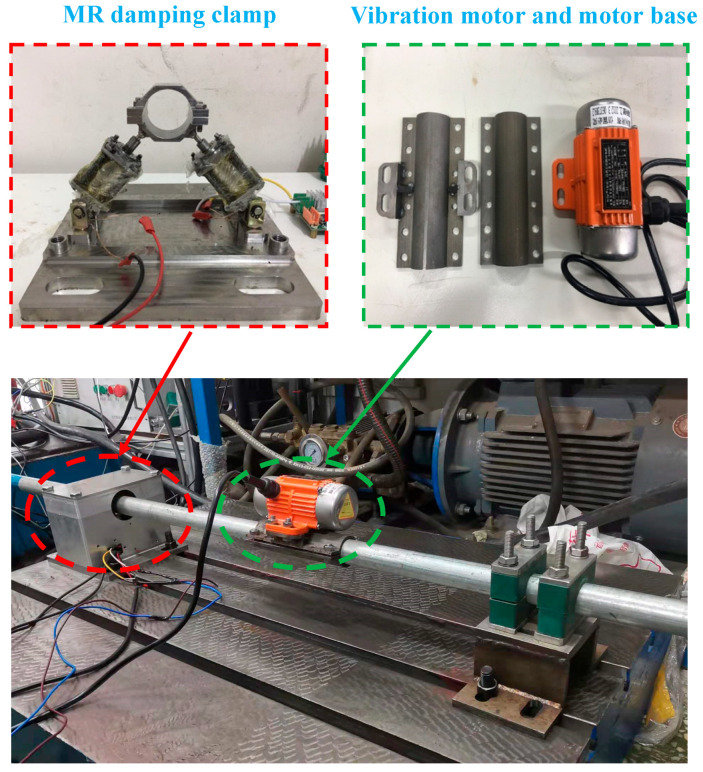
Experiment device structure.

**Figure 12 micromachines-13-00531-f012:**
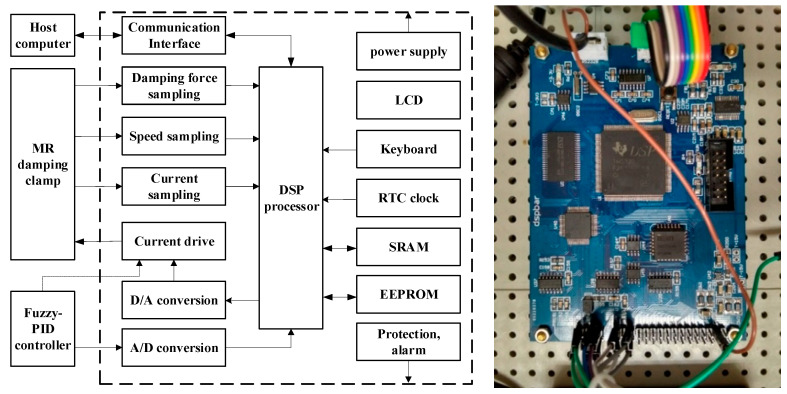
Structure diagram of pipeline vibration control system.

**Figure 13 micromachines-13-00531-f013:**
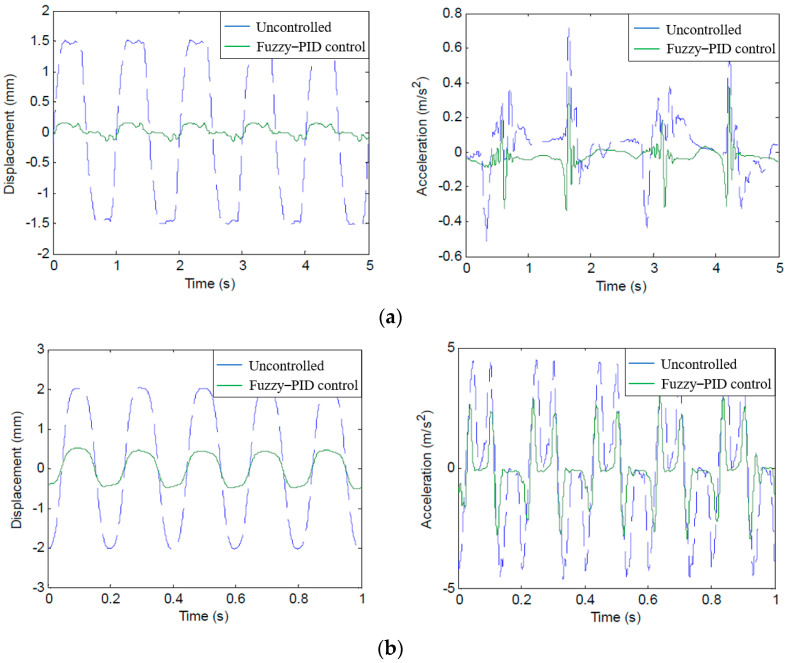
Pipeline vibration response under fuzzy–PID control. (**a**) Excitation frequency 1 Hz. (**b**) Excitation frequency 5 Hz. (**c**) Excitation frequency 10 Hz.

**Table 1 micromachines-13-00531-t001:** Main design parameters of the MR damping clamp.

Content	Parameter	Content	Parameter
Piston rod radius	4 mm	Shell thickness	3 mm
Damping gap width	0.8 mm	Number of field turns	350
Trunk width	8 mm	Working stroke	±10 mm
Trunk depth	4 mm	Spring stiffness	240 N/mm
Piston flange width	3 mm	Range of working current	0–2.0 A

**Table 2 micromachines-13-00531-t002:** Fuzzy rules of ∆*k_p_*.

F/u/f	NB	NM	NS	ZE	PS	PM	PB
NB	PB	PM	PM	PM	PM	PS	PS
NM	PB	PM	PM	PM	PS	PS	ZE
NS	PM	PM	PS	ZE	ZE	ZE	PS
ZE	PS	PS	ZE	ZE	ZE	PS	PS
PS	PS	ZE	ZE	ZE	PS	PS	PM
PM	ZE	PS	PS	PM	PM	PM	PB
PB	PS	PS	NS	PM	PM	PB	PB

**Table 3 micromachines-13-00531-t003:** Fuzzy rules of ∆*k_i_*.

F/u/f	NB	NM	NS	ZE	PS	PM	PB
NB	NB	NB	NM	NM	NS	NS	ZE
NM	NB	NM	NM	NS	NS	ZE	ZE
NS	NM	NM	NS	ZE	ZE	ZE	PS
ZE	NS	NS	ZE	ZE	ZE	PS	PS
PS	NS	AE	ZE	ZE	PS	PS	PM
PM	ZE	ZE	PS	PS	PM	PM	PB
PB	ZE	PS	PS	PM	PM	PB	PB

**Table 4 micromachines-13-00531-t004:** Fuzzy rules of ∆*k_d_*.

F/u/f	NB	NM	NS	ZE	PS	PM	PB
NB	PB	PM	PM	PS	PS	ZE	ZE
NM	PM	PM	PS	PS	ZE	ZE	ZE
NS	PM	PS	PS	ZE	ZE	ZE	NS
ZE	PS	PS	ZE	ZE	ZE	NS	NS
PS	PS	ZE	ZE	ZE	NS	NS	NM
PM	ZE	ZE	ZE	NS	NS	NM	NM
PB	ZE	ZE	NS	NS	NM	NM	NB

**Table 5 micromachines-13-00531-t005:** Vibration response of the pipeline with fuzzy–PID control.

Frequency (Hz)	Displacement				Acceleration			
	Maximum Value (mm)	Root Mean Square (mm)	VAL (dB)	Attenuation Rate	Maximum Value (m/s^2^)	Root Mean Square (m/s^2^)	VAL (dB)	Attenuation Rate
1	0.1534	0.0879	23.4504	93.28%	0.3837	0.0864	6.0256	50.03%
2	0.1764	0.1320	20.9159	91%	0.4268	0.0924	15.9341	84.03%
3	0.2889	0.1872	18.3291	87.88%	1.5022	0.3991	8.8125	63.74%
4	0.2464	0.1659	19.6381	89.57%	1.5670	0.4677	11.2885	72.74%
5	0.2481	0.3691	13.0984	77.87%	3.3275	1.1832	6.6085	53.27%
6	0.2869	0.2113	18.0531	87.49%	2.6603	1.0532	9.9406	68.16%
7	0.3554	0.3116	15.0427	82.30%	2.4870	0.7274	15.4165	83.05%
8	0.2518	0.3034	14.5912	81.36%	3.5160	1.3785	10.8271	71.25%
9	0.2523	0.1194	22.1668	92.21%	3.2596	1.1516	13.6211	79.16%
10	0.3138	0.1963	17.8309	87.16%	3.3683	1.4250	12.1991	75.45%

**Table 6 micromachines-13-00531-t006:** Background of vibration control technologies of hydraulic pipeline.

Work	Control Technologies	Frequency Range (Hz)	Vibration Reduction Ratio (%)
[16]	Phononic crystals of support clamp	600~1000 Hz	50
[17]	Pounding tuned mass damper	1.404–1.953	80.8
[18]	Constrained layer damping treatment	30–50 Hz	32.7
[19]	Tuned Mass Particle Damper	5.66–10.31	38.76
[20]	Cantilever beam tuned mass dampers	110–130	10.2
[21]	Tuned mass dampers with frequency adjustable	66–84	65.6
[22]	Active Control of in-series and By-pass Structures	40–120 Hz	35.2

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
