# Peer review of "Pipeline Vibration Control Using Magnetorheological Damping Clamps under Fuzzy–PID Control Algorithm"

_micromachines, 2022, doi:10.3390/mi13040531_

Round 1
Reviewer 1 Report
1) Please avoid repetition in the section of the introduction
2) Please elaborate the section of results with more appropriate discussions. In my view, this paper is worth publishing but not in the current format. Appropriate results discussions must be added to the paper.
Reviewer 2 Report
Controlling the vibrations in pipeline is a challenging work. For the time being, many passive vibration control methods are proposed in the open literatures. There are few research literatures on active vibration control of pipeline, so the author's work is meaningful for this field. This paper investigated the semi-active vibration control method of pipeline system based on MR damping clamp. However a few improvements are needed. This paper can be accepted if the following problems have been revised.
(1) A concise and factual abstract is required. The abstract should state briefly the purpose of the research, the principal results and major conclusions.
(2) The authors are suggested to add some latest references to give a better perspective of recent vibration control of pipeline system.
(3) In section 4.1, the acceleration sensor and displacement sensor are not clear in the experiment system. It is suggested to add the description of test bench. Such as pump speed, fluid pressure, sensor position, etc.
(4) The conclusion is too long to be shortened to specific points. Moreover, try to suggest some possible extensions to the work. This is very important and needs to be included in conclusions as it sets the correct premise for future directions.
(5) The results show that MR damping clamp effectively suppress the vibration response of 0.5Hz~2.5Hz. For engineering application, the pipeline system are often suffering from the serious vibration by pump pressure fluctuation excitation with wide frequency (5Hz~2000Hz). It is suggested to add the discussion.
(6) The language of the manuscript needs major elaboration. Please elaborate the text by getting assistance from a native English speaker.
Reviewer 3 Report
The authors present the article entitled “Pipeline vibration control using magnetorheological damping 2 clamps under fuzzy-PID control algorithm”. However, in its current form, it is not possible to extend my recommendation for publication by the following concerns:
The manuscript is found with a lot of grammatical and typographical errors. The authors are suggested to go through the manuscript thoroughly and get it to proofread for grammatical and typographical errors.
References are not presented in the required style. Please check the instruction for authors.
Please, at the end of the introduction, including the structure of the manuscript.
Lines 48 to 58: There are two “at present…” sentences. What is the actual paradigm of the damping pipe clamp?
Lines 49-51: this sentence is ambiguous.
Lines 62-64: Reference is missing.
From my perspective, I recommend synthesizing the Introduction section to provide a detailed study to find out the motive and novelty of the present work by doing an extensive comparison of the presented literature in the manuscript.
Vectorize the figures in order to see the details.
Equations 10 and 11 should be presented first in section 3. Section 3 should be named methodology and subsection 4.2 must be presented in a section called Results.
Include a table that compares the findings of the work vs the already reported in state of the art.
The manuscript must be checked by a native English speaker.
Define all your acronyms as DSP.
The vibration issues of line 43 can be justified by considering the following references:
A new methodology for a retrofitted self-tuned controller with open-source fpga; Non-linear regression models with vibration amplitude optimization algorithms in a microturbine; Frequency and time-frequency analysis of cutting force and vibration signals for tool condition monitoring
Include dots at the end of the captions of sentences as “Figure 1. Structure and schematic of magnetorheological damping clamp”
In the last section, please add future works.
Round 2
Reviewer 1 Report
My comments were addressed fully and it is accepted.
Reviewer 2 Report
This paper has been revised according to comments. I recommend this paper can be accepted.
Reviewer 3 Report
My comments have been addressed.